# Dimensions of Cellulose Nanocrystals from Cotton and Bacterial Cellulose: Comparison of Microscopy and Scattering Techniques

**DOI:** 10.3390/nano14050455

**Published:** 2024-02-29

**Authors:** Vladimir Grachev, Olivier Deschaume, Peter R. Lang, Minne Paul Lettinga, Carmen Bartic, Wim Thielemans

**Affiliations:** 1Sustainable Materials Lab, Department of Chemical Engineering, KU Leuven, Campus Kulak Kortrijk, Etienne Sabbelaan 53, 8500 Kortrijk, Belgium; vladimir.grachev@kuleuven.be; 2Laboratory for Soft Matter Physics and Biophysics, Department of Physics and Astronomy, KU Leuven, Celestijnenlaan 200D Box 2416, 3001 Leuven, Belgium; olivier.deschaume@kuleuven.be (O.D.); pavlik.lettinga@kuleuven.be (M.P.L.); carmen.bartic@kuleuven.be (C.B.); 3Institute for Biomacromolecular Systems and Processes Group (IBI-4), Forschungszentrum Jülich, Wilhelm-Johnen-Straße, 52428 Jülich, Germany; p.lang@fz-juelich.de

**Keywords:** sulfated cellulose nanocrystals, carboxylated cellulose nanocrystals, dimension determination, AFM, SAXS, DLS, SLS, cotton, bacterial cellulose

## Abstract

Different microscopy and scattering methods used in the literature to determine the dimensions of cellulose nanocrystals derived from cotton and bacterial cellulose were compared to investigate potential bias and discrepancies. Atomic force microscopy (AFM), small-angle X-ray scattering (SAXS), depolarized dynamic light scattering (DDLS), and static light scattering (SLS) were compared. The lengths, widths, and heights of the particles and their respective distributions were determined by AFM. In agreement with previous work, the CNCs were found to have a ribbon-like shape, regardless of the source of cellulose or the surface functional groups. Tip broadening and agglomeration of the particles during deposition cause AFM-derived lateral dimensions to be systematically larger those obtained from SAXS measurements. The radius of gyration determined by SLS showed a good correlation with the dimensions obtained by AFM. The hydrodynamic lateral dimensions determined by DDLS were found to have the same magnitude as either the width or height obtained from the other techniques; however, the precision of DDLS was limited due to the mismatch between the cylindrical model and the actual shape of the CNCs, and to constraints in the fitting procedure. Therefore, the combination of AFM and SAXS, or microscopy and small-angle scattering, is recommended for the most accurate determination of CNC dimensions.

## 1. Introduction

Cellulose nanocrystals (CNCs) are particles with a highly anisometric shape that belong to a class of nanomaterials obtained through the controlled acid hydrolysis of naturally occurring cellulose. The CNCs can be obtained from a wide variety of cellulose sources, such as wood [1,2,3,4], algae [5,6], cotton [7,8], tunicates [9,10], bacterial cellulose [11,12], and plant waste biomass [13]. The CNCs can possess various functional groups on their surface. The functional groups can be formed as part of the acid hydrolysis, e.g., sulfate [14], phosphate [15], or carboxylate [16], or can be introduced by targeted surface modifications of the surface hydroxyl groups, such as the oxidation of primary alcohol groups into carboxylic acid groups [17,18]. The negatively charged CNCs can form stable colloidal suspensions in water owing to the electrostatic repulsion between the charged groups [19]. Their high uniaxial mechanical strength and Young’s modulus imply the suitability of CNCs as reinforcing components in nanocomposites [20,21]. More recently, CNCs have also gained interest in other applications, using them as stabilizers of Pickering emulsions [22], as components of supercapacitors [23], as catalyst support [24], and as flocculants [25].

CNCs are especially interesting, though, because they readily form nematic [11] and chiral nematic [26,27,28,29] liquid crystalline phases at elevated concentrations. The concentration where the transition from the orientationally disordered isotropic phase to the orientationally ordered nematic phase takes place, referred to as the isotropic–nematic binodal point, was theoretically predicted by Onsager [30] for a system of thin, rigid rods and reworked by Stroobants et al. [31] for a system of charged rod-like particles. According to these theories, the isotropic–nematic binodal point is inversely proportional to the aspect ratio of the particles, which describes the length-to-diameter ratio in the case of rod-like particles. However, the CNCs were shown to not have a strictly rod-like shape, but a ribbon-like shape, which was characterized by three dimensions, length *L*, width *b*, and height *h*, with *L* >> *b* >> *h* [27,32,33,34,35]. Therefore, it is important to have knowledge of these dimensions when rationalizing where the isotropic–nematic phase transition takes place and to compare the behavior of different samples.

Various microscopy and scattering techniques have been employed to determine the dimensions of CNCs. The most common techniques used for the size characterization of CNCs are atomic force microscopy (AFM) [36,37,38] and transmission electron microscopy (TEM) [11,39,40,41,42], which allow us to visualize the particles and to determine all three dimensions of the particles together with their respective distributions when measuring a sufficiently large number of particles. Furthermore, AFM provides information on surface morphological and mechanical properties of the crystals [36,43]. While AFM has been shown to lead to the accurate determination of the length of the CNCs, the lateral resolution of this method is reduced because of tip broadening artefacts [44] and due to the aggregation of the particles during deposition. TEM, on the other hand, provides a good nanometric lateral resolution, allowing us to screen a large population of the particles rapidly [45,46]; but, it is a challenge to determine the height of the particles with TEM since TEM images are two-dimensional projections of the objects along the incident beam direction, and the low electron density of the CNCs requires the use of staining methods [46]. Electron microscope tomography has also been shown to reconstruct the 3D structures in aerogels [47] and chiral twisting of pristine CNCs [48,49] and chitin nanocrystals [50], and therefore can be used to determine both the width and height of the particles.

Various scattering techniques, such as small-angle neutron scattering (SANS), small-angle X-ray scattering (SAXS), and static light scattering (SLS), allow for the determination of the dimensions of the particles in situ and in a dispersed state from a subnanometric scale up to a few microns [34,51]. SAXS [35,52,53] and SANS [54,55,56,57] have been employed to accurately determine the lateral dimensions of the CNCs using parallelepiped form factor models, and SAXS has also been used as a complementary technique to AFM to determine the lateral dimensions of CNCs [58]. However, the accuracy of length determination by small-angle scattering is reported to be low [54] and is considered practically unreliable for nanocellulose systems [35] as it depends mostly on measurements at the lowest angles, which are prone to errors related to primary beam corrections. Alternatively, polarized and depolarized dynamic light scattering (DLS and DDLS) have been used to determine the hydrodynamic dimensions of anisometric particles [59], as well as the dynamic properties of CNCs, such as translational and rotational diffusion coefficients [60,61,62], whereas static light scattering (SLS) allows us to determine the radius of gyration of the particles [34] and to estimate the equivalent size of pristine and aggregated CNCs [63].

Although complementary methods, such as SAXS, have been recently used in combination with microscopy techniques for a more precise characterization of the dimensions of the CNCs [35,39,58,64,65], there is a gap in understanding the suitability of light scattering techniques to accurately characterize the particle size. A systematic comparison of all these techniques is required to investigate instrumental bias and to explain the differences between the different approaches.

The objective of this paper is therefore twofold. First, we want to identify the most reliable experiments (or combination of experiments) to obtain a complete view of the particle size distribution when comparing AFM, SAXS, DDLS, and multi-angle SLS. Second, we apply this toolbox to the dimensions of five different samples of cellulose nanocrystal suspensions, obtained from cotton wool and bacterial cellulose. We first introduce the preparation method of these different systems and the technical details of our experimental toolbox. Then, we present the results for all the systems in the order indicated above. Finally, we discuss what are the best options for carrying out the characterization of these systems.

## 2. Materials and Methods

### 2.1. Materials

Cotton wool was purchased from Hartmann AG (Heidenheim, Germany). Nata de coco was purchased from Tijn’s Toko (Amsterdam, The Netherlands). Sulfuric acid (95%), 0.1 M of hydrochloric acid standard solution, methanol (99%), Dowex Marathon H-form, and Amberlite MB6113 mixed-bed ion exchange resins were obtained from Fisher Scientific (Hampton, NH, USA). Hydrochloric acid (37%) and sodium hypochlorite solution (14% Cl_2_) were acquired from VWR (Leuven, Belgium). Spectra/Por type-4 (MWCO 16–18 kDa) dialysis tubing membranes were obtained from SpectrumLabs (Rancho Dominguez, CA, USA). Sodium bromide (99%) was acquired from Carl Roth GmbH (Karlsruhe, Germany). Sodium hydroxide (pellets, analysis grade) and 2,2,6,6-tetramethylpiperidyloxyl (TEMPO) were purchased from Merck (Darmstadt, Germany). All products were used as received.

### 2.2. Preparation of CNC Suspensions

Sulfated cellulose nanocrystals were extracted from cotton wool by sulfuric acid hydrolysis [26]. Fifty grams of cotton wool were hydrolyzed with 450 mL of 64 wt% sulfuric acid at 45 °C for 40–45 min, repeatedly centrifuged at 8000 rpm at 4 °C with the intermittent decantation of the supernatant and redispersion of the sediment in deionized water until the sediment was no longer separable from the supernatant, and dialyzed against deionized water until the conductivity of the effluent water was not higher than 5 μS/cm. The dialyzed CNC suspension was sonicated using a high-performance sonifier (Branson Digital Sonifier 250, Emerson Electric, Ferguson, MO, USA) and filtered through a sintered glass filter (pore size 2). The CNC suspension was thereafter treated with an H^+^-form ion exchange resin (previously rinsed with deionized water until we achieved a neutral pH and conductivity not higher than 5 μS/cm). The suspension was then neutralized with freshly prepared 0.1 M sodium hydroxide solution.

Carboxylated cellulose nanocrystals were obtained by the hydrolysis of the cotton wool with 4 M of hydrochloric acid [14] and subsequent TEMPO-mediated oxidation [17]. Twenty-five grams of cotton wool were hydrolyzed for 4 h at 80 °C, repeatedly centrifuged and dialyzed following the procedure described above. The dialyzed CNC suspension was treated with mixed-bed ion exchange resin overnight and was thereafter sonicated using a high-performance sonifier (Branson Digital Sonifier 250). The concentration of the CNCs obtained by HCl hydrolysis was determined by thermogravimetric analysis (TGA) on a Netzsch TG209F3 instrument. For TEMPO-mediated oxidation, the CNC suspension was mixed with solid TEMPO (30 mg/g solid CNCs) and sodium bromide (300 mg/g solid CNCs) until complete dissolution. Sodium hypochlorite (14% Cl_2_, 2.5 mL/g solid CNCs) was added to the suspension, and the reaction was performed for 40 min at room temperature with the pH maintained at 11. The oxidation was subsequently quenched with methanol (2 mL/g solid CNCs). The suspension was then neutralized with 0.1 M of hydrochloric acid, dialyzed against deionized water, sonicated on a high-performance sonifier, filtered through a pore-size 2 sintered glass filter, and treated with an H^+^-form ion exchange resin following the procedure described for sulfate CNCs. The suspension was neutralized with a freshly prepared 0.1 M sodium hydroxide solution.

Mixed sulfated-carboxylated cotton cellulose nanocrystals were prepared by the sulfuric acid hydrolysis of cotton wool and subsequent TEMPO-mediated oxidation, following the procedures described above.

Bacterial cellulose sulfated nanocrystals were obtained by the sulfuric acid hydrolysis of pre-treated bacterial cellulose from nata de coco [66]. Raw nata de coco cubes were treated with 1 M of sodium hydroxide solution overnight, then washed with deionized water 3 times, and homogenized with a kitchen hand blender. The resulting pulp was repeatedly washed with deionized water and filtered through a sintered glass filter (pore size 2) until we achieved neutrality of the filtrate. The neutralized pulp was freeze-dried on a Christ freeze-drier at −50 °C and at 1.3 mbar. The dried material was then hydrolyzed with 60 wt% sulfuric acid (100 mL of acid per 1 g of dried bacterial cellulose), repeatedly centrifuged at 13,000 rpm with the intermittent removal of the supernatant and redispersion of the sediment in deionized water, sonicated on a high-performance sonifier (Branson Digital Sonifier 250), and dialyzed against deionized water until the conductivity of the effluent water was not higher than 5 μS/cm. The dialyzed suspension was sonicated and filtered. The CNC suspension was thereafter treated with an H^+^-form ion exchange resin (previously rinsed with deionized water until we achieved a neutral pH and conductivity not higher than 5 μS/cm). The suspension was neutralized with a freshly prepared 0.1 M sodium hydroxide solution.

### 2.3. Experimental Techniques for the Size Characterization of CNC Suspensions

Atomic force microscopy (AFM) experiments were carried out on an Agilent 5500 system with an MSNL-F triangular cantilever (frequency = 110–120 kHz, *k* = 0.6 N/m, tip radius = 2–12 nm). The diluted (0.01–0.10 wt%) CNC suspensions were deposited on freshly cleaned silicon substrates modified with polyallylamine hydrochloride. A total of 5 μL of the suspension was incubated on the substrates for 1 min, rinsed with Milli-Q water, and dried using a nitrogen gas flow. The topographical imaging was performed in an intermittent contact mode. Analyses of the length, width, and height of the particles were carried out using Gwyddion image analysis software [67] on 5 μm × 5 μm images with at least 100 particles.

Small-angle X-ray scattering (SAXS) measurements were performed on a Xenocs Xeuss 2.0 C laboratory beamline equipped with an ultra-low dispersion copper K_α_ X-ray source and DECTRIS Eiger 1 M detector. The X-ray beam was collimated to a circular beam with diameter 0.5 mm. The samples (0.1 wt% cotton sulfated and sulfated-carboxylated CNCs, 0.05 wt% cotton carboxylated CNCs, and 0.02 wt% bacterial sulfated CNCs) were loaded into borosilicate glass capillaries with a diameter of 1.0 mm and wall thickness of 0.01 mm, and sealed with low-temperature glue. The entire beam path was brought to high (0.1–0.3 mbar) vacuum prior to the measurements. The sample-to-detector distance was calibrated before the data acquisition using silver (I) behenate. The measurements were performed three times for each sample. The resulting 2D data were reduced to 1D data using Foxtrot (Xenocs/Synchrotron du Soleil) software with absolute intensity scaling according to the measured intensity of the transmitted beam. A blank experiment performed with deionized water was subtracted from the resulting 1D data. Fitting of the subtracted 1D data was carried out using SasView 5.0.5 using the parallelepiped model for the particle form factor with width and height polydispersities. The scattering length density of cellulose was fixed at 14.5 Å^−2^.

Depolarized dynamic light scattering (DDLS) and static light scattering (SLS) measurements were performed simultaneously on a an ALV SLS/DLS setup. A Spectra-Physics 532 nm laser with a 150 mW maximum power was used as a light source. The incident laser beam was polarized vertically to the scattering plane with a Glan–Thompson prism polarizer (B. Halle, extinction rate 10^−6^), while the scattered light was polarized parallel to the scattering plane with a second identical Glan–Thompson prism. The second polarizer was adjusted to minimize the transmission of the primary beam at zero angle. The setup was equipped with an ALV CGS-3 goniometer and an ALV 5000/E Multiple Tau correlator. The sample cell was positioned in a toluene bath to match the refractive index of the glass while the temperature was maintained at 25 °C by a Julabo F12 thermostat throughout all the experiments. The scattering angle was changed with an ALV/LSE 5000/II controller. The measurements of CNCs obtained from cotton wool were carried out at scattering angles between 16° and 30° with angular steps of 2°, between 30° and 60° in steps of 5°, and between 60° and 120° in steps of 10°. The measurements of CNCs obtained from bacterial cellulose were performed at scattering angles between 20° and 60° in steps of 5° and between 60° and 120° in steps of 10°. For each scattering angle, one autocorrelation function was acquired. The fitting of the autocorrelation functions was carried out using the software SimpleSim, which was coded in-house allowing the simultaneous fit of correlation functions recorded at different scattering angles with either the model for translational and rotational diffusions of blunt-end cylinders or stretched exponential functions. Static light scattering intensity was acquired at each scattering angle simultaneously with the DDLS autocorrelation functions. The radius of gyration of the particles, *R_g_*, was determined from Guinier plots of the scattered intensities applying linear least-squares fits to the intensities at small scattering vectors.

### 2.4. Data Analysis

The AFM measurements were performed to determine the length, width, and height of the CNCs with corresponding size distributions. The length, *L*, and the width, *b*, were determined manually by measuring the distances determined from the size of the cross-section in two perpendicular directions, while the height, *h*, was determined by extracting profiles along arbitrary lines, with the threshold set at 10% of the maximum height. The dimensions were fitted with a lognormal distribution in accordance with previous reports [39,61]. The lognormal distribution is defined as:(1)$$y\;=\;\frac{A}{\sigma x}\exp(-\frac{1}{2}{\left(\frac{\ln x-\;\mu }{\sigma }\right)}^{2})$$
where *A* is the normalization factor, and *μ* and *σ* are the mean value and the standard deviation of the logarithm of the measured variable, *x*, respectively. The mean value of the measured variable itself is defined as:(2)$$x_{mean}=\exp(\mu +\frac{\sigma^{2}}{2})$$
with the *σ* parameter describing the width of the underlying normal distribution and the polydispersity. The standard deviation is calculated as the square root of the variance of the measured variable:(3)$$\Delta x=\sqrt{\exp\left(2\mu +\sigma^{2}\right)*(\exp\left(\sigma^{2}\right)-1})$$

To fit the SAXS data, we used the parallelepiped model with three independent dimensions, *L*, *b*, and *h*, as indicated above, where *I*, as a function of the scattering vector, *q*, is given by:(4)$$I\left(q\right)=\frac{scale}{Lbh}{\left(Lbh*(\rho_{particle}-\rho_{solvent})\right)}^{2}*P\left(q,a,c\right)+background$$
where *ρ_particle_* and *ρ_solvent_* are the scattering length densities of the particles and of the solvent, respectively, $a\;=\;\frac{h}{b}$ and $c\;=\;\frac{L}{b}$ [34,68]. The form factor, *P*(*q*, *a*, *c*), is defined as:(5)$$P\left(q,a,c\right)=\int_{0}^{1}\phi_{q}(\mu \sqrt{1-\sigma^{2}},\;a){\left[\frac{\sin(\frac{\mu c\sigma }{2})}{\frac{\mu c\sigma }{2}}\right]}^{2}du$$
where
(6)$$\phi_{q}\left(\mu ,a\right)=\int_{0}^{1}{\left\{\frac{\sin(\frac{\mu }{2}\cos(\frac{\pi u}{2}))}{\frac{\mu }{2}\cos(\frac{\pi u}{2})}\frac{\sin(\frac{\mu a}{2}\sin\left(\frac{\pi u}{2}\right))}{\frac{\mu a}{2}\sin(\frac{\pi u}{2})}\right\}}^{2}du$$
and *μ* = *qb*. To reduce the number of parameters of the fit, the mean length of the parallelepiped was fixed to the length determined by AFM as it does not impact the shape of the scattering curves in the accessible *q*-range, while the length polydispersity was neglected. The fitting was performed in two steps. In the first step, the width and lognormal width distribution were determined. In the second step, the width and its distribution were fixed to the values obtained during the first step, and the height and its lognormal distribution were fitted to the data. In order to prevent the contribution of a structure factor to the SAXS measurements, the concentrations (in terms of volume fraction) of the diluted suspensions for the SAXS experiments were chosen such that they were always less than 50% of the overlap concentration, *c**, defined as:(7)$$c^{*}=\frac{V_{particle}}{V_{sphere}}=\frac{\pi LX^{2}}{\frac{4}{3}\pi {\left(\frac{L}{2}\right)}^{3}}=\frac{6}{\pi }*{\left(\frac{X}{L}\right)}^{2}$$
where *V_particle_* is the volume of the parallelepiped, *V_sphere_* is the volume of a hypothetical sphere with the diameter of a parallelepiped length, and *X* is the lateral dimension of the parallelepiped, either the width or height. The *c** values, presented in Table 1, were calculated with *X* equal to the height.

To obtain the CNC dimensions from DDLS, fifteen to twenty autocorrelation functions recorded at different scattering angles in the VH configuration were fitted simultaneously to the blunt-end cylinder model [69] with a monodisperse hydrodynamic cross-section dimension, *d_hd_*, and the length polydispersity parameter, *σ_L_*. The mean length, $\langle L\rangle$, was fixed as determined by AFM. The length polydispersity was determined assuming a lognormal distribution. For angles small enough to warrant *q**$\langle L\rangle$ < 5, where *q* is the scattering vector, the correlation function of the scattered intensity for monodisperse rods is given by:(8)$$g_{2}\left(t\right)-1=\exp(-\left(2D_{tr}q^{2}+12D_{rot}\right)t)$$

To obtain averaged translational and rotational diffusion coefficients, the experimental autocorrelation functions were fitted to the single stretched exponential model:(9)$$g_{2}\left(t,q\right)-1=A\;exp\left\{-{\left(\frac{t}{C}\right)}^{\beta }\right\}+B$$
where *β* is the stretching parameter, and the mean relaxation frequency, $\langle \Gamma \rangle$, is calculated using *β* and the parameter *C* as:(10)$$\langle \Gamma \rangle =\frac{\beta }{C}*\frac{1}{\Gamma (\frac{1}{\beta })}$$
with Γ on the right-hand side of Equation (10) denoting the gamma function. The mean relaxation frequency, $\langle \Gamma \rangle$, is then plotted as a function of *q*^2^ to determine the averaged translational, $\langle D_{tr}\rangle$, and rotational, $\langle D_{rot}\rangle$, diffusion coefficients according to:(11)$$\langle \Gamma \rangle =2\langle D_{tr}\rangle q^{2}+12\langle D_{rot}\rangle$$

The experimentally obtained diffusion coefficients, $\langle D_{tr}\rangle$ and $\langle D_{rot}\rangle$, were compared with theoretical diffusion coefficients. The dependence of theoretical translational *D_tr_* and rotational *D_rot_* coefficients on the geometric parameters of the rod are assumed to follow:(12)$$D_{tr}=\frac{kT}{3\pi \eta L}*(\ln\left(\frac{\langle L\rangle }{d_{hd}}\right)+0.312+0.565*\frac{d_{hd}}{\langle L\rangle }-0.100*{\left(\frac{d_{hd}}{\langle L\rangle }\right)}^{2})$$
and
(13)$$D_{rot}=\frac{3kT}{\pi \eta L^{3}}*(\ln\left(\frac{\langle L\rangle }{d_{hd}}\right)-0.662+0.917*\frac{d_{hd}}{\langle L\rangle }-0.050*{\left(\frac{d_{hd}}{\langle L\rangle }\right)}^{2})$$
as given by Tirado and García de la Torre [70] for blunt-end cylinders.

For comparison with the experimental data, additional autocorrelation functions were calculated by applying Equations (8), (12) and (13), integrating over a lognormal distribution of the length, and assuming that:
-Average length $\langle L\rangle$ is fixed as obtained from AFM;-Hydrodynamic cross-section dimension $d_{hd}$ is fixed at either the width or the height as obtained from AFM;-Length polydispersity *σ* is fixed at either 0.1, 0.3, 0.5, or 0.8.

The calculated autocorrelation functions were similarly fitted with the single stretched exponential model, and the average theoretical diffusion coefficients $\langle D_{tr}\rangle$ and $\langle D_{rot}\rangle$ were calculated following Equation (11). The concentrations of the CNC suspensions used for DDLS are shown in Table 1.

The radius of gyration, *R_g_*, was obtained by fitting the logarithm of SLS intensity with the Guinier equation:(14)$$\ln I=\ln I_{0}-q^{2}*(\frac{R_{g}^{2}}{3})$$
in the small-angle range, where *q* ∗ *R_g_* ≤ 1.

## 3. Results and Discussion

### 3.1. Atomic Force Microscopy (AFM)

Figure 1 displays the AFM images of the sulfated cotton CNCs. From these images we determined the mean length, the mean width, and the mean height to be 189 ± 60 nm, 42 ± 11 nm, and 6.7 ± 1.2 nm, respectively. Therefore, given the differences between the length and width, and width and height, the particles have a ribbon-like shape. The *σ_L_* parameter of 0.18–0.31 indicates a moderately polydisperse suspension. The dimensions of sulfated-carboxylated cotton CNCs (Figure 2) are similar to the sulfated CNCs as shown in Table 2. However, the polydispersity is higher with the σ parameter in the range of 0.27–0.38. The polydispersity parameters and lateral dimensions of carboxylated cotton CNCs (Figure 3) are similar to the sulfated-carboxylated CNCs. Both sulfated-carboxylated and carboxylated CNCs have a ribbon-like geometry. However, the carboxylated cotton CNCs are significantly longer, as their mean length determined by AFM (277 ± 77 nm) is 47% larger than the mean length of sulfated CNCs and 36% larger than the mean length of sulfated-carboxylated CNCs. This difference in the mean length is attributed to the different methodologies used to prepare carboxylated CNCs. The hydrolysis of cellulose with sulfuric acid takes place under harsher conditions than the hydrolysis with hydrochloric acid because the concentration of sulfuric acid (10.1 M) is higher than the concentration of hydrochloric acid (4 M). Hence, the hydrolysis of cellulose with hydrochloric acid can result in less degradation of dislocated regions of cellulose, and the resulting particles are longer than those obtained by sulfuric acid treatment. TEMPO-mediated oxidation after sulfation, on the other hand, did not result in a significant change in dimensions as shown by the similar sizes obtained for sulfated and sulfated-carboxylated CNCs.

Furthermore, two different batches of sulfated CNCs from bacterial cellulose were investigated (Figure 4). The bacterial CNCs possess a ribbon-like shape similar to cotton CNCs, which is reflected in the large length-to-width and width-to-height ratios (Table 2). The mean dimensions of the two batches were found to be consistent, although a significant difference was observed in the mean height and in the height polydispersity. Both batches of the sulfated bacterial CNCs have a high polydispersity in length. The differences in the mean dimensions and polydispersity can arise from variations in the raw material or from small differences in the hydrolysis conditions, thus in the residual water content of the freeze-dried bacterial cellulose before acid hydrolysis, since an increased water content decreases the efficiency of hydrolysis.

### 3.2. Small-Angle X-ray Scattering (SAXS)

SAXS was employed to determine the lateral dimensions of the particles in a diluted suspension. The parallelepiped model provides a good (R^2^ > 0.995) fit for the SAXS data of the cotton CNCs with the various functional groups and different batches of sulfated bacterial CNCs, as shown in Figure 5 (fit residuals shown in Appendix A). The ribbon-like geometry of cotton and bacterial CNCs as obtained by AFM was confirmed by the high values of width-to-height ratios, presented in Table 3.

### 3.3. Depolarized Dynamic Light Scattering (DDLS)

DDLS was used to determine the hydrodynamic cross-section dimensions (CSDs), $d_{hd}$, of the CNCs and polydispersity in length. The autocorrelation functions of cotton CNCs with various functional groups were fitted to the cylindrical model with a good quality of fit, indicated by R^2^ > 0.998 (Figure 6, fit residuals shown in Appendix A). The hydrodynamic CSDs of sulfated and carboxylated CNCs, presented in Table 4, show a good correlation with the width as obtained by SAXS, although they are smaller than the widths obtained by AFM. On the other hand, the hydrodynamic CSD of the CNCs bearing both functional groups were estimated at 9.6 nm, which was more than two-times smaller than the hydrodynamic CSDs of CNCs with only one functional group, and was closer to the mean height of the particles. The difference in the hydrodynamic cross-section dimension between sulfated and sulfated-carboxylated CNCs could have been caused by the increase in the surface charge after TEMPO-mediated oxidation, which in turn resulted in a decrease in the Debye length [62].

We used Equation (9) to fit the experimental autocorrelation functions with the single stretched exponential model, and Equations (10) and (11) to determine the experimental translational diffusion coefficient, $\langle D_{tr}\rangle$, and rotational diffusion coefficient, $\langle D_{rot}\rangle$. Thereafter, to determine which dimensions yielded the theoretical autocorrelation function that most closely matched the experimental function, we calculated the theoretical autocorrelation function using Equation (8) with the theoretical diffusion coefficients determined from the model by Tirado and García de la Torre [68] (Equations (12) and (13)) using the width, $\langle b\rangle$, the height, $\langle h\rangle$, or their mean arithmetic, $\frac{\langle b\rangle +\langle h\rangle }{2}$, as the input parameter for $d_{hd}$ in the blunt-end cylinder model, while systematically varying the length polydispersity, *σ_L_*. Then, the average theoretical diffusion coefficients, $\langle D_{tr}\rangle$ and $\langle D_{rot}\rangle$, were obtained from the calculated autocorrelation functions, following Equations (10) and (11), and were compared with the experimentally determined diffusion coefficients. The diffusion coefficients are shown in Appendix A.

The fit of the $\langle \Gamma \rangle$ vs. *q*^2^ plots of cotton CNCs (Figure 7) exhibit a systematic mismatch when the hydrodynamic CSD is assumed to be equal to the AFM-derived width, and experimentally determined translational diffusion coefficients are significantly higher than the calculated ones. The mismatch was reduced when the hydrodynamic CSD was set to the AFM-derived height or to the mean arithmetic of the width and the height, $\frac{\langle b\rangle +\langle h\rangle }{2}$, for the theoretical diffusion coefficient calculations. Introducing length polydispersity decreased the mismatch further, and experimental and calculated $\langle \Gamma \rangle$ vs. *q*^2^ plots matched best at length polydispersities of 0.5 and 0.8. The best match of experimental and calculated diffusion coefficients, shown in Appendix A, was observed for a length polydispersity of 0.5, with $d_{hd}$ being equal to the height for sulfated-carboxylated CNCs and to $\frac{\langle b\rangle +\langle h\rangle }{2}$ for CNCs, with only one surface functional group. This correlates with a higher length polydispersity obtained by DDLS in comparison with AFM, as presented in Table 4.

The autocorrelation functions of bacterial CNCs, shown in Figure 8, are noisier at small relaxation times due to a lower scattering intensity, but can also be well-fitted. The hydrodynamic CSDs of the bacterial CNCs, shown in Table 4, are similar to the mean heights from AFM, and the length polydispersity is significantly higher than the length polydispersity derived from AFM. The $\langle \Gamma \rangle$ vs. *q^2^* plots (Figure 9) and the comparison between experimental and calculated diffusion coefficients (Appendix A) show the same results, with experimental and calculated $\langle \Gamma \rangle$ vs. *q^2^* plots, translational and rotational diffusion coefficients matching best for a length polydispersity of 0.8 and for a hydrodynamic CSD equal to the AFM-determined height.

### 3.4. Static Light Scattering (SLS)

Multi-angle SLS allows us to determine the radius of gyration of the particles, (*R_g_*), which is proportional to the average distance of each point of the particle relative to its center of mass. Therefore, *R_g_* is a measure of the particle size related to its mass distribution. The Guinier plots of the CNCs, shown in Figure 10, were fitted to a straight line in the scattering angle range of 16–40° with a good quality of fit (R^2^ > 0.95). The values of *R_g_* for cotton CNCs are shown in Table 5. It should be noted that, for bacterial CNCs, which are shown to have larger lengths, the SLS has a stricter limitation on the scattering angle range, and scattering angles larger than 30° are not compliant with the *q* ∗ *R_g_* ≤ 1 rule.

Furthermore, the radii of gyration obtained from SLS were compared with the *R_g_* values calculated from the length, width, and height values from AFM. For a parallelepiped with a length *L*, a width *b*, and a height *h*:(15)$$R_{g}=\frac{1}{12}\sqrt{L^{2}+b^{2}+h^{2}}$$
and the standard deviation of *R_g_* was calculated from standard deviations of the corresponding AFM-derived parameters through the error propagation formula:(16)$$\Delta R_{g}=\frac{L\Delta L+b\Delta b+h\Delta h}{12R_{g}}$$

As shown in Table 5, the *R_g_* values determined with SLS are close to the values calculated from dimensions measured by AFM, with the difference between *R_g_* values obtained from the two techniques not exceeding 20%. The standard deviations of the *R_g_* values from SLS are significantly lower than the standard deviations of the AFM-derived radii of gyration, because the reflect the fit uncertainty but not the actual size distribution.

### 3.5. Discussion

The length of cotton CNCs bearing various functional groups and the length of two batches of sulfated bacterial CNCs were determined with AFM. However, the resolution of AFM to determine the lateral dimension is reduced as compared to the resolution in length, mainly because of tip broadening. The tip diameter, *d_tip_*, is added to the true width of the particle in the images, resulting in an apparent width, *b**, determined from the images given by:(17)$$b^{*}=b+d_{tip}$$

Because the tip diameter can vary significantly, this results in a systematic error in width, the magnitude of which varies with each tip. Furthermore, this effect also depends on the tip shape and the tip height. Therefore, it is difficult to accurately correct the width for this broadening effect without the use of reference structures, and the apparent width determined by AFM is biased towards larger values. Moreover, the lateral aggregation of the particles during deposition also results in an increase in width, which is reflected in the appearance of deformed stacks in the AFM images (Figure 1, Figure 2, Figure 3 and Figure 4).

The lateral dimensions derived from SAXS were found to be systematically smaller than the lateral dimensions determined by AFM, despite the large standard deviations, as shown in Table 2 and Table 3. The systematic difference in width was in the range of 10–20 nm for all analyzed samples, whereas the systematic difference in height was in the range of 2–4 nm for most of the samples. In principle, SAXS should provide larger dimensions than AFM because SAXS measures intensity-averaged dimensions, *X_SAXS_*, as given by:(18)$$X_{SAXS}=\frac{\sum_{k}X_{k}^{6}N_{k}}{\sum_{k}X_{k}^{5}N_{k}}$$
where *N_k_* is the number of particles with the dimension *X_k_*, and AFM measures number-averaged dimensions, *X_AFM_*, as given by:(19)$$X_{AFM}=\frac{\sum_{k}X_{k}N_{k}}{\sum_{k}N_{k}}$$

As scattered intensity is proportional to the 6th power of the hydrodynamic size of the particle [71], the contribution of the larger particles in the intensity-averaged dimensions is more significant than in the number-averaged dimensions. In our study, however, SAXS provides smaller width and height values compared to AFM, which contradicts this approach. This contradiction may be caused, in the order of decreasing likelihood, by AFM tip broadening, because the differences in width and height have the same magnitude as the typical AFM tip radius observed for the probe used in similar measurement conditions, by large polydispersity in dimensions, or by particle stacking.

The determination of CNC dimensions by DDLS was complicated because of the polydispersity in length, which was found to be significantly higher than the analogous parameter derived from AFM, as presented in Table 4. The polydispersity in length from DDLS for bacterial CNCs was so significant that the standard deviation of length, calculated from Equation (3), was larger than the mean value of the length itself. It is critical to mention that the fitting procedure assumes a cylindrical shape of the particles, which means that the mismatch between the cylindrical model and the actual ribbon-like shape results in a systematic error of the technique. Furthermore, the dramatic increase in polydispersity in length can be attributed to the limitations of the fitting algorithm to polydispersity of only one size parameter; so, in this case the error in length also includes errors in the cross-section dimensions. A fitting procedure with polydispersity in two size parameters, e.g., length and hydrodynamic CSD, could theoretically improve the correlation between the length polydispersities from DDLS and from AFM, although it would be computationally demanding.

Static light scattering, which was used to determine the radius of gyration of the particles, did not provide additional information about the actual dimensions and polydispersities. Therefore, SLS can be used as a complementary technique to microscopy (AFM, TEM) and small-angle scattering (SAXS, SANS).

A comparison of all the techniques, for accurate the determination of dimensions of CNCs, shows that a combination of atomic force microscopy and small-angle X-ray scattering is recommended because microscopy allows us to determine the mean length and length polydispersity accurately and to visualize the particles, while small-angle scattering allows a more accurate determination of lateral dimensions and their respective polydispersities.

## 4. Conclusions

The dimensions and size distributions of sulfated, carboxylated, and sulfated-carboxylated cotton CNCs and two batches of bacterial sulfated CNCs were determined with atomic force microscopy, small-angle X-ray scattering, depolarized dynamic light scattering, and static light scattering. AFM was found to be the most suitable technique to determine the length of the particles and the length distribution, however the determination of lateral dimensions of the CNCs with AFM was complicated due to the tip broadening and aggregation of the particles in the process of deposition. A more accurate determination of CNC width, height, and the corresponding distributions can be obtained using SAXS, using a parallelepiped cross-section model. The combined data from AFM and SAXS reveal that the CNCs have a ribbon-like shape in line with the previous reports. Depolarized DLS can be used to determine the hydrodynamic dimensions of the CNCs, but the accuracy of this technique for the determination of CNC dimensions is limited because of the ribbon-like shape of the particles and limitations of the fitting procedure. Furthermore, the radius of gyration of the CNCs can be determined with SLS at small scattering angles in good agreement to the AFM-derived dimensions. The combination of AFM and SAXS, or electron microscopy and small-angle scattering, is therefore recommended for the most accurate determination of dimensions of the CNCs. With the increasing accuracy and improved resolution of characterization techniques, future work should continue these comparisons, in addition to the inclusion of additional techniques, e.g., (cryo-)electron microscopy, and the improvement of models for fits used for dynamic light scattering and X-ray scattering. In addition, the large width-to-height ratio found for CNCs in this work and that of others is also something to be investigated as it is much larger than is expected and indicates the lengthwise alignment and aggregation of CNCs under characterization conditions.

## Figures and Tables

**Figure 1 nanomaterials-14-00455-f001:**
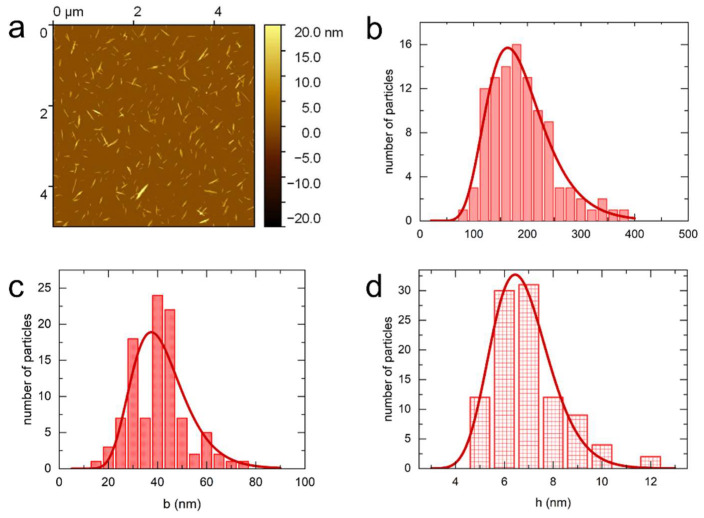
AFM characteristic image (**a**) and the corresponding distribution in length (**b**), width (**c**), and height (**d**) for sulfated cotton CNCs.

**Figure 2 nanomaterials-14-00455-f002:**
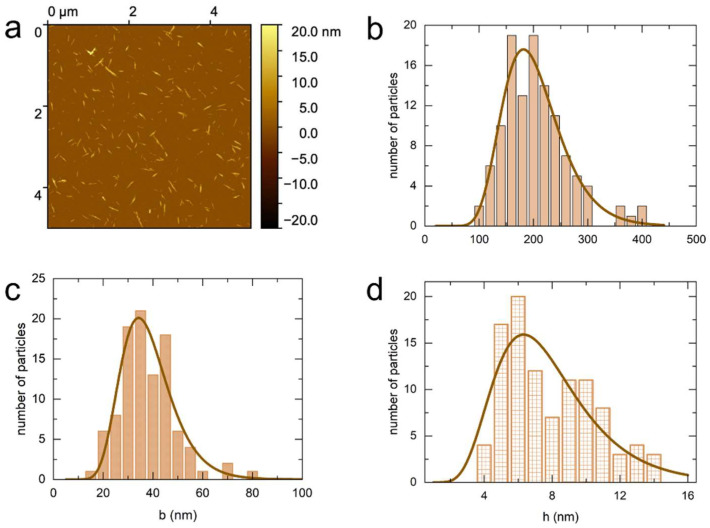
AFM characteristic image (**a**) and the corresponding distribution in length (**b**), width (**c**), and height (**d**) for sulfated-carboxylated cotton CNCs.

**Figure 3 nanomaterials-14-00455-f003:**
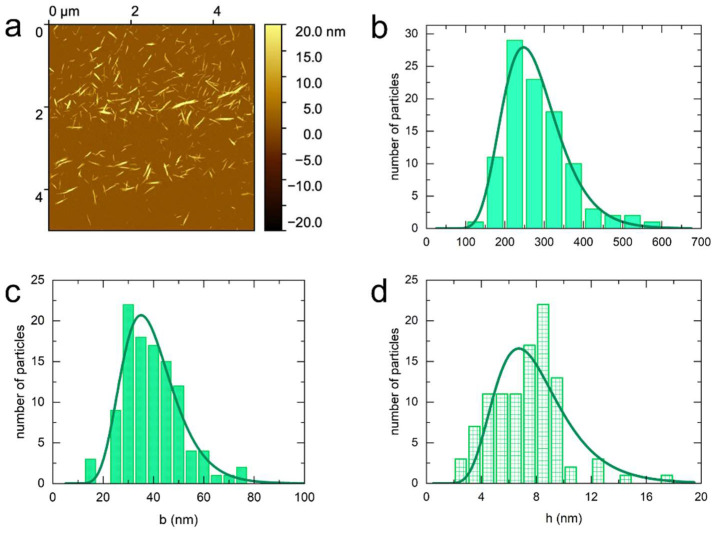
AFM characteristic image (**a**) and the corresponding distribution in length (**b**), width (**c**), and height (**d**) for carboxylated cotton CNCs.

**Figure 4 nanomaterials-14-00455-f004:**
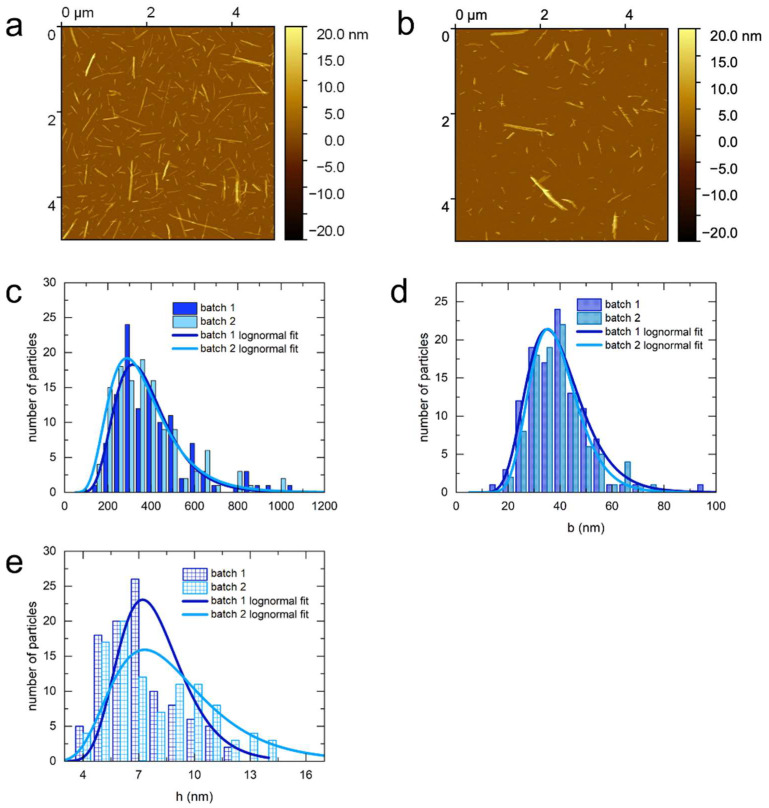
AFM characteristic images (**a**,**b**) and the corresponding distributions in length (**c**), width (**d**), and height for two batches of sulfated bacterial CNCs (**e**).

**Figure 5 nanomaterials-14-00455-f005:**
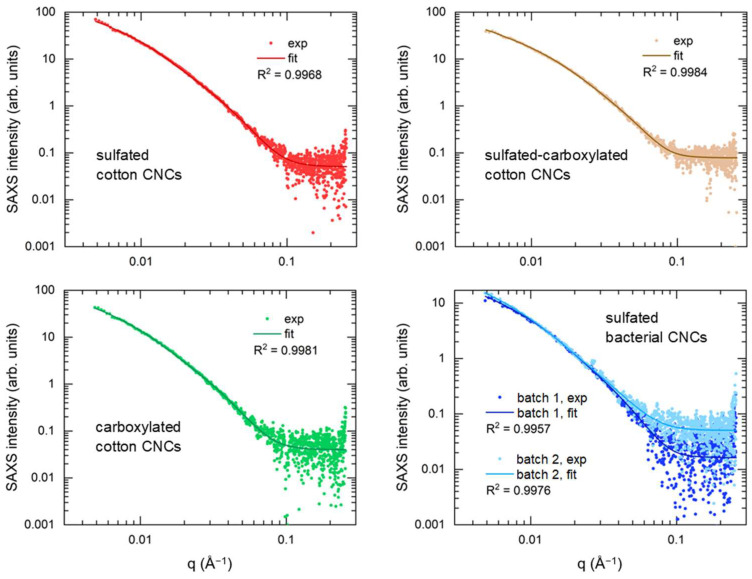
SAXS intensity plots of sulfated, sulfated-carboxylated, and carboxylated cotton CNCs, and two batches of sulfated bacterial CNCs.

**Figure 6 nanomaterials-14-00455-f006:**
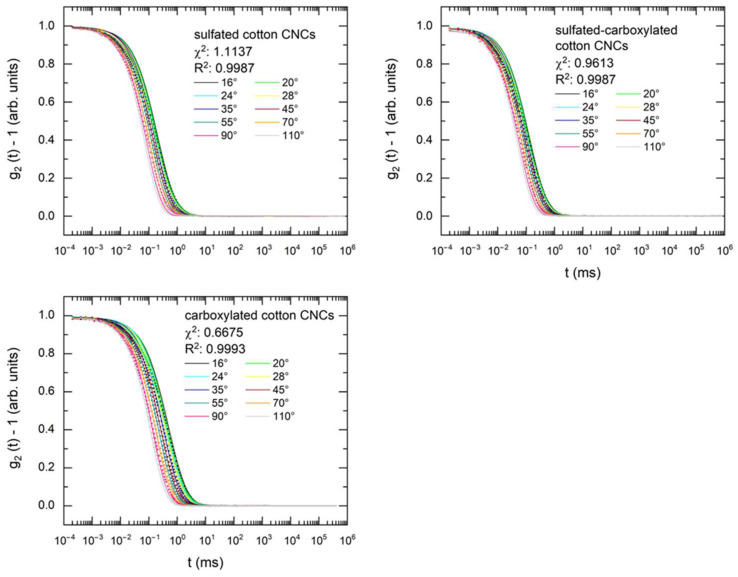
Experimental and fitted DDLS autocorrelation functions at different scattering angles for sulfated, sulfated-carboxylated, and carboxylated cotton CNCs.

**Figure 7 nanomaterials-14-00455-f007:**
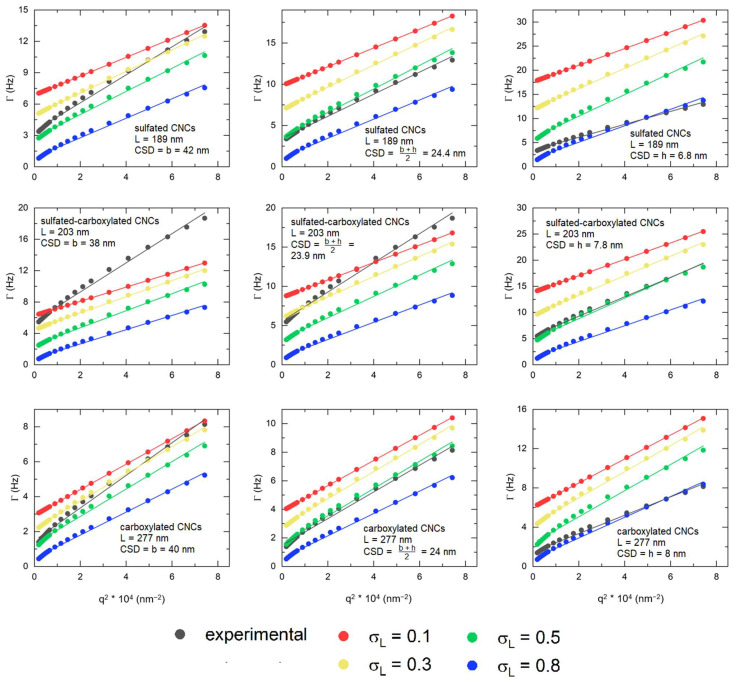
$\langle \Gamma \rangle$ vs. *q^2^* plots of experimental (black) and calculated autocorrelation functions for cotton CNCs. Calculated autocorrelation functions were obtained by setting the length polydispersity to 0.1 (red), 0.3 (yellow), 0.5 (green), and 0.8 (blue).

**Figure 8 nanomaterials-14-00455-f008:**
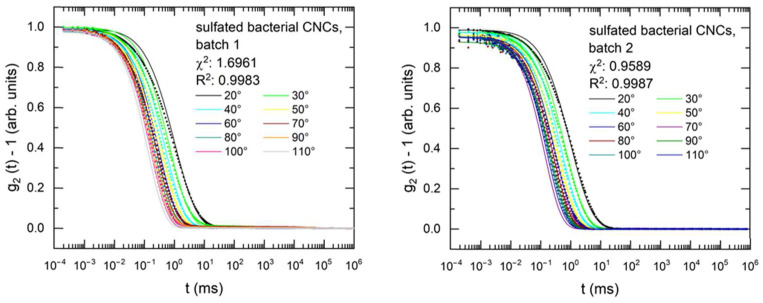
Experimental and fitted DDLS autocorrelation functions at different scattering angles for bacterial CNCs.

**Figure 9 nanomaterials-14-00455-f009:**
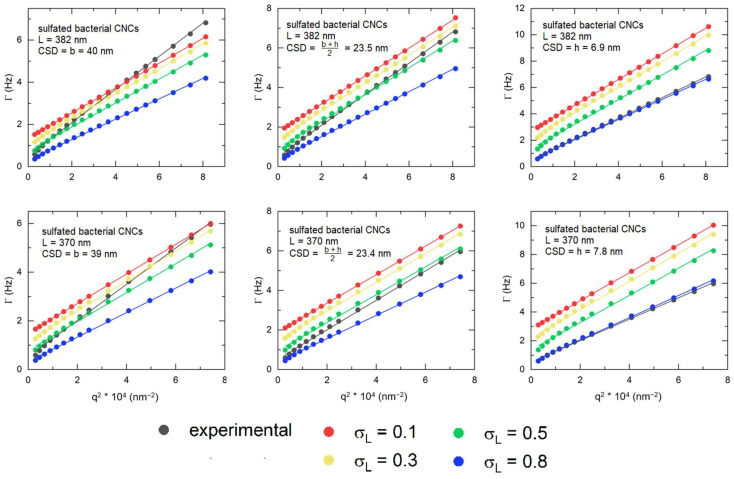
$\langle \Gamma \rangle$ vs. *q*^2^ plots of experimental (black) and calculated autocorrelation functions for sulfated bacterial CNCs. Calculated autocorrelation functions were obtained by setting the length polydispersity to 0.1 (red), 0.3 (yellow), 0.5 (green), and 0.8 (blue).

**Figure 10 nanomaterials-14-00455-f010:**
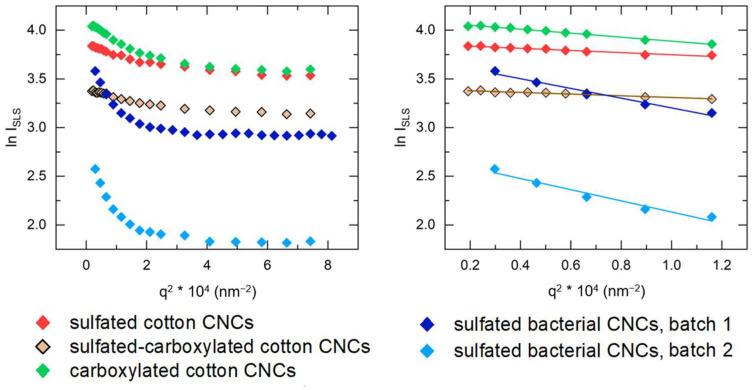
Logarithm of static light scattering intensity ln(*I_SLS_*) vs. *q*^2^ plots of diluted CNC suspensions at scattering angle ranges of 16–130° (**left**) and 16–40° (**right**).

**Table 1 nanomaterials-14-00455-t001:** Calculated values of the overlap concentration of CNCs, *c**, the concentrations used for SAXS measurements, *c_SAXS_*, and the concentrations used for DDLS measurements, *c_DDLS_*.

Sample	*c**, vol%	*c_SAXS_*, vol%	*c_DDLS_*, vol%
sulfated cotton CNCs	0.25	0.10	0.05
sulfated-carboxylated cotton CNCs	0.28	0.10	0.05
carboxylated cotton CNCs	0.16	0.05	0.05
sulfated bacterial CNCs, 1st batch	0.06	0.02	0.01
sulfated bacterial CNCs, 2nd batch	0.09	0.02	0.01

**Table 2 nanomaterials-14-00455-t002:** Dimensions (mean length $\langle L\rangle$, mean width $\langle b\rangle$, and mean height $\langle h\rangle$) and polydispersity parameters (polydispersity *σ_L_* in length, *σ_b_* in width, and *σ_h_* in height) of the cotton and bacterial CNCs determined by AFM.

Parameter	Sulfated Cotton CNCs	Sulfated-Carboxylated Cotton CNCs	Carboxylated Cotton CNCs	Sulfated Bacterial CNCs, 1st Batch	Sulfated Bacterial CNCs, 2nd Batch
$\langle L\rangle$, nm	189	203	277	382	370
*σ_L_*	0.31 ± 0.02	0.27 ± 0.02	0.27 ± 0.01	0.36 ± 0.03	0.41 ± 0.03
$\langle b\rangle$, nm	42	38	40	40	39
*σ_b_*	0.27 ± 0.05	0.28 ± 0.02	0.28 ± 0.02	0.29 ± 0.02	0.25 ± 0.01
$\langle h\rangle$, nm	6.8	7.8	8.0	6.9	7.8
*σ_h_*	0.18 ± 0.01	0.38 ± 0.05	0.34 ± 0.05	0.27 ± 0.03	0.38 ± 0.05
$\langle L\rangle$:$\langle b\rangle$ ratio	4.5	5.3	6.9	9.6	9.5
$\langle b\rangle$:$\langle h\rangle$ ratio	6.2	4.9	5.0	5.8	5.0

**Table 3 nanomaterials-14-00455-t003:** Lateral dimensions (mean width $\langle b\rangle$ and width polydispersity *σ_b_*, mean height $\langle h\rangle$ and height polydispersity *σ_h_*) and width-to-height ratios ($\langle b\rangle$:$\langle h\rangle$) of cotton and bacterial CNCs determined by SAXS.

Parameter	Sulfated Cotton CNCs	Sulfated-Carboxylated Cotton CNCs	Carboxylated Cotton CNCs	Sulfated Bacterial CNCs, 1st Batch	Sulfated Bacterial CNCs, 2nd Batch
$\langle b\rangle$, nm	28	20	31	26	29
*σ_b_*	0.28 ± 0.02	0.34 ± 0.03	0.29 ± 0.03	0.29 ± 0.05	0.35 ± 0.06
$\langle h\rangle$, nm	4.5	5.0	5.1	5.0	3.9
*σ_h_*	0.42 ± 0.02	0.29 ± 0.01	0.42 ± 0.01	0.29 ± 0.08	0.44 ± 0.02
$\langle b\rangle$:$\langle h\rangle$ ratio	6.2	4.0	6.1	5.2	7.4

**Table 4 nanomaterials-14-00455-t004:** Comparison of the hydrodynamic cross-section dimensions, $d_{hd}$, of cotton and bacterial CNCs determined by DDLS, lateral dimensions (mean width, $\langle b\rangle$, and mean height, $\langle h\rangle$) determined by AFM, length polydispersities, *σ_L_*, determined by DDLS and AFM.

Parameter	Sulfated Cotton CNCs	Sulfated-Carboxylated Cotton CNCs	Carboxylated Cotton CNCs	Sulfated Bacterial CNCs, 1st Batch	Sulfated Bacterial CNCs, 2nd Batch
$d_{hd}$, nm	28.4	9.6	25.0	5.7	10.0
$\langle b\rangle$ (AFM), nm	42	38	40	40	39
$\langle h\rangle$ (AFM), nm	6.8	7.8	8.0	6.9	7.8
*σ_L_* (DDLS)	0.50	0.45	0.53	0.81	0.78
*σ_L_* (AFM)	0.31	0.27	0.27	0.36	0.41

**Table 5 nanomaterials-14-00455-t005:** Radius of gyration *R_g_* (nm) of cotton and bacterial CNCs determined by AFM and SLS.

Method	Sulfated Cotton CNCs	Sulfated-Carboxylated Cotton CNCs	Carboxylated Cotton CNCs	Sulfated Bacterial CNCs, 1st Batch	Sulfated Bacterial CNCs, 2nd Batch
AFM	56 ± 18	60 ± 17	81 ± 23	111 ± 41	107 ± 46
SLS	57 ± 2	51 ± 2	78 ± 1	122 ± 6	131 ± 7

## Data Availability

Raw data are available from DOI-referenced RDR via https://doi.org/10.48804/WOGERN.

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
