# Peer review of "Dimensions of Cellulose Nanocrystals from Cotton and Bacterial Cellulose: Comparison of Microscopy and Scattering Techniques"

_nanomaterials, 2024, doi:10.3390/nano14050455_

Round 1

Reviewer 1 Report

Comments and Suggestions for Authors

Authors performed a comparative investigation on structural characteristics of various kinds of cellulose nanocrystals, employing AFM, light and X-ray scattering. The results show the usefulness of the combination of the above mentioned methods to obtain reliable experimental data on these nanosystems. The purpose of the study is stated clearly, the methodology is well explained and the analysis is performed on a high level. The paper can be suitable for publication after some improvements.

1. Authors can relate their findings and data analysis approach to that used in recent SAXS and SANS investigations of cellulose nanofibrils

Valencia et al., https://doi.org/10.1039/D0NR02888F
Rossetti et al., https://doi.org/10.1007/s10570-023-05058-2
Smyslov et al., https://doi.org/10.3390/biomimetics8070520
Ivanova et al., https://doi.org/10.3390/ma13092087

2. Characteristic results, such as nanocrystal dimensions, should be put in the Abstract and/or Conclusions.

3. The English may be improved, in particular, to avoid phrases such as:
line 138: A blank water experiment

Author Response

Authors performed a comparative investigation on structural characteristics of various kinds of cellulose nanocrystals, employing AFM, light and X-ray scattering. The results show the usefulness of the combination of the above mentioned methods to obtain reliable experimental data on these nanosystems. The purpose of the study is stated clearly, the methodology is well explained and the analysis is performed on a high level. The paper can be suitable for publication after some improvements. 
Thank you very much for this positive initial assessment.

  1. Authors can relate their findings and data analysis approach to that used in recent SAXS and SANS investigations of cellulose nanofibrils

    Valencia et al., https://doi.org/10.1039/D0NR02888F 
    Rossetti et al., https://doi.org/10.1007/s10570-023-05058-2
    Smyslov et al., https://doi.org/10.3390/biomimetics8070520
    Ivanova et al., https://doi.org/10.3390/ma13092087

These manuscripts all relate to work with cellulose nanofibrils (CNFs). The procedure described in this manuscript was optimized only for cellulose nanocrystals (CNCs). A trustworthy size determination of CNFs will require different techniques. For example, length could be difficult to be determined with AFM, and the usability of DLS is limited because of the higher viscosity of CNF dispersions. Therefore, adding references related to CNFs, in addition to CNCs, into the paper devoted solely to CNCs will result in a confusion which we cannot justify justified. Based on this, we decided that we could not, in good faith, add these manuscripts to this paper.

2. Characteristic results, such as nanocrystal dimensions, should be put in the Abstract and/or Conclusions.

We agree with the reviewer that this could be useful, and we did try to do it. However, we have five samples, three dimensions and two complementary dimensions, and three widths of dimension distributions (polydispersity). It became very confusing and the abstract and conclusion lost the real message of the work, i.e., that one needs the combination of AFM with scattering is needed for accurate determination of all three dimensions and distributions. The actual values for the dimensions are less relevant since they will vary with cellulose source and hydrolysis conditions, however, correct determination of these dimensions for each batch one makes is really important for comparison, and that is were this paper provides a contribution.

3. The English may be improved, in particular, to avoid phrases such as:
line 138: A blank water experiment 

This particular sentence was rephrased. We have also gone through the paper and made it better readable.

Reviewer 2 Report

Comments and Suggestions for Authors

The manuscript titled “Dimensions of cellulose nanocrystals from cotton and bacterial cellulose: comparison of microscopy and scattering techniques” by Grachev, V.; et al. is a scientific work where the authors assess the morphological dimensions of cellulose nanocrystals (CNCs) from five different sources (sultated cotton, sulfated-carboxylated cotton, carboxylated cotton, and two batches of bacterial cellulose) obtained by acid hydrolysis. This study attempts to fill the gap between the data acquire by single-molecule techniques as atomic force microscopy (AFM) and complementary light scattering tools like small angle X-ray scattering (SAXS), depolarized dynamic light scattering (DDLS) and static light scattering (SLS), respectively. These light scattering techniques allow the precise determination of the length and widht dimensions preventing the detrimental inherent broadening tip-sample effects observed during AFM scanning measurements. This study is interesting and the manuscrip is generally well-written.

However, it exists some points that need to be addressed (please, see them below detailed point-by-point). The most relevant outcomes remarked by the authors can contribute in the growth of many fields like the design and development of plant-based materials which require an extensive and accurate characterization of their morphological dimensions, among other physical properties. For this reason, I will recommend the present scientific manuscript for further publication in the Nanomaterials once all the below described suggestions will be properly fixed.

Here, there exists some points that must be covered in order to improve the scientific quality of the manuscript paper:

1) KEYWORDS. The authors should add the most relevant terms related to this work in the keyword list.

2) INTRODUCTION. “AFM provides information on surface topography, morphology and mechanical properties” (lines 51-52). Here, it lacks a relevant reference related to the obtention of mechanical parameters by atomic force microscopy [1].

[1] Magazzù, A.; et al. Investigation of Soft Matter Nanomechanics by Atomic Force Microscopy and Optical Tweezers: A Comprehensive Review. Nanomaterials 2023, 13, 963. https://doi.org/10.3390/nano13060963.

3) “While AFM (…) due to aggregation of the particles during deposition” (lines 52-54). Does this aggegation effect appear during TEM sample preparation (on grids) or it is only common for the AFM measurements? Then, the authors showed isolated CNCs in this work (as seen in Figs. 1-4). Some good practice tips should be provided in this regard in order to avoid this detrimental effect.

4) MATERIALS AND METHODS. “Preparation of the CNC suspensions” (lines 90-123). What were the reaction yields for the CNCs obtention coming from the five different sources?

5) “Carboxylated cellulose (…) TEMPO-mediation oxidation” (lines 99-100). Please, the authors should define the full-name of the TEMPO chemical compound. Then, the abbreviation should be placed between brackets. This comment should be taken into account for the rest of the main manuscript body text.

6) “Analysis (…) Gwyddion image analysis software (…) particles” (lines 129-130). The following bibliography reference should be taken into account [2].

[2] Nečas, D.; et al. Gwyddion: an open-source software for SPM data analysis. Open Phys. 2012, 10, 181-188. https://doi.org/10.2478/s11534-011-0096-2.

7) “Depolarized dynamic light scatterin (DDLS) (…) scattered intensities applying linear least-squares fits at small scattering vectors” (lines 141-155). How many measurements (population size) were carried out for each CNCs examined sample? Did the authors measure the samples in triplicate as SAXS experiments?

8) RESULTS AND DICUSSION. “Fig.1 displays (…) water content decreases the efficiency of hydrolysis” (lines 224-259). Here, the authors present the gathered results obtained for the five examined samples (which are summarized in Table 2). It may be opportune to compare these gathered data to the morphological dimensions of CNCs coming from other sources like softwood pulp (length: 67.0 ± 3.0 nm and height: 3.4 ± 0.1 nm)  [3], Ramie plant fibers (length: 180.9 ± 12.5 nm and height: 6.3 ± 0.4 nm) [4] or coconut gels (length: 196.8 nm and height: 7.0 ± 1.5 nm) [5].

[3] Chen, M.; et al. AFM characterization of cellulose nanocrystal height and width using internal calibration standards. Cellulose 2021, 28, 1933-1946. https://doi.org/10.1007/s10570-021-03678-0.

[4] Marcuello, C.; et al. Langmuir-Blodgett Procedure to Precisely Control the Coverage of Functionalized AFM Cantilevers for SMFS Measurements: Application with Cellulose Nanocrystals. Langmuir 2018, 34, 9376-9386. https://doi.org/10.1021/acs.langmuir.8b01892.

[5] Yurtsever, A.; et al. Molecular insights on the crystalline cellulose-water interfaces via three-dimensional atomic force microscopy. Sci. Adv. 2022, 8, eabq0160. https://doi.org/10.1126/sciadv.abq0160.

9) Fig. 1, panel a (line 240). Why does the vertical bar render negative topographic values (the background is settled at -20.0 nm)? Same comment for the Fig.2, panel a (line 243), Fig.3, panel a (line 246) and Fig. 4, panels a and b (line 260).

10) SUPPLEMENTARY INFORMATION. Table S1. Please, the authors should homogenize the significant figures of the data displayed in this table. Same comment for the Tables S2-S5.

11) DISCUSSION. This section perfectly explained the findings of this work with comprehensive explanations. No actions are requested from the authors.

12) This section clearly states the most relevant outcomes outlined in this work. The authors should add a brief statement about the future action lines to pursue this research.

Comments on the Quality of English Language

The manuscript is generally well-written albeit it may be advisable if the authors could recheck it in order to polish some final details susceptible to be improved.
